# Insights from *Syzygium aromaticum* Essential Oil: Encapsulation, Characterization, and Antioxidant Activity

**DOI:** 10.3390/ph17050599

**Published:** 2024-05-08

**Authors:** Naianny L. O. N. Mergulhão, Laisa C. G. Bulhões, Valdemir C. Silva, Ilza F. B. Duarte, Irinaldo D. Basílio-Júnior, Johnnatan D. Freitas, Adeildo J. Oliveira, Marília O. F. Goulart, Círia V. Barbosa, João X. Araújo-Júnior

**Affiliations:** 1Institute of Pharmaceutical Sciences, Federal University of Alagoas (UFAL), Maceió 57072-900, Brazil; naianny.nascimento@esenfar.ufal.br (N.L.O.N.M.); laisacarolina@hotmail.com (L.C.G.B.); irinaldo.junior@icf.ufal.br (I.D.B.-J.); ciria.barbosa@icf.ufal.br (C.V.B.); 2Program of the Northeast Biotechnology Network (RENORBIO), Institute of Chemistry and Biotechnology, Federal University of Alagoas (UFAL), Maceió 57072-900, Brazil; valdemir.silva@estacio.br; 3Estácio de Alagoas Faculty, Maceió 57035-225, Brazil; 4Department of Food Chemistry, Federal Institute of Alagoas, Maceió 57020-600, Brazil; johnnatandf@gmail.com; 5Department of Exact Sciences, Federal University of Alagoas, Arapiraca 57309-005, Brazil; adeildojro@gmail.com

**Keywords:** alginate, eugenol, clove essential oil, freeze drying, ionic gelation

## Abstract

Alginate encapsulates loaded with clove essential oil (CEO) were prepared by ionic gelation, with subsequent freeze-drying. The objective of the present work was to develop a product with the ability to protect CEO against its easy volatility and oxidation. The following techniques were used to characterize the formulations: eugenol release, degree of swelling, GC/MS, TGA/DSC, and SEM. The alginate solution (1.0%) containing different concentrations of CEO (LF1: 1.0%; LF2: 0.5%; LF3: 0.1%) was dropped into a 3.0% CaCl_2_ solution. After lyophilization, the encapsulated samples were wrinkled and rigid, with high encapsulation power (LF3: 76.9% ± 0.5). Three chemical components were identified: eugenol (the major one), caryophyllene, and humulene. The antioxidant power (LF1: DPPH IC_50_ 18.1 µg mL^−1^) was consistent with the phenol content (LF1: 172.2 mg GAE g^−1^). The encapsulated ones were thermally stable, as shown by analysis of FTIR peaks, eugenol molecular structure was kept unaltered. The degree of swelling was 19.2% (PBS). The release of eugenol (92.5%) in the PBS solution was faster than in the acidic medium. It was concluded that the low-cost technology used allows the maintenance of the content and characteristics of CEO in the three concentrations tested, offering a basis for further research with essential oil encapsulates.

## 1. Introduction

Technologies such as encapsulation systems provide an alternate means of maintaining the characteristics of the products that are trapped in matrixes. They include methods for enhancing the physical-chemical stability and simplifying the transportation, delivery, and concentration of active ingredients [1,2]. When encasing delicate materials, one of the most popular encapsulation techniques is ionic gelation. This technique is inexpensive and simple to replicate. Its applications are also expanding and promising because it does not call for the use of organic solvents or high temperatures [3,4]. It is crucial to employ a drying procedure since the encapsulated product’s high-water content reduces its thermal stability.

Encasing essential oils helps stop oxidation and the loss of volatile chemicals in the food industry. Encapsulation enhances shelf life, masks flavors, and increases nutritional content [5,6]. Active encapsulation is used in the pharmaceutical industry to ensure patient safety and efficacy by guarding against environmental elements such as light, oxygen, and humidity, that can degrade encapsulated components, which are still limiting factors in the conservation of these products [7,8,9,10,11].

An example of low chemical stability is found in eugenol, the major component of cloves, which is susceptible to oxidation and many biochemical interactions [8]. Clove essential oil (CEO) is known to be a highly promising compound for the pharmaceutical sector, having several activities including antibacterial [12], antifungal [13], antiparasitic [14], antioxidant [15,16], anti-inflammatory [17], and anesthetic [18] effects. Therefore, it is pertinent to intensify research involving the encapsulation of essential oils to maintain their natural characteristics and their chemical content [10].

In addition to being one of the best methods for preventing degradation under adverse environmental circumstances, encapsulation can also be utilized to increase the lifespan of products [10,11]. The food and pharmaceutical sectors are always very interested in goods that are improved by the encapsulation of essential oils to preserve and protect the active content. This field of study has a great deal of scientific significance.

Ionic gelation encapsulates of alginate are naturally biodegradable, perhaps more sustainable, and less expensive than other forms of encapsulates reported in the literature, such as those based on synthetic polymers [19,20]. Alginates are highly common polymers in the food business; they are utilized as thickening, gel-forming, stabilizing, and active ingredient carriers [21]. Spirulina protein may be encapsulated by alginate, which makes it easier to create food with different protein sources [22]. Alginate is used in this ionic gelation encapsulation technology to trap many probiotics, improve their viability, and raise the likelihood that *L. acidophilus* [23,24], *L. reuteri* [25], *L. casei* [23,26,27], and *Lactococcus lactis* [28] microbes survive adverse digestive tract circumstances. Alginate encapsulates containing linseed oil extract [29] and *Satureja hortensis* essential oil [30] were created using ionic gelation and have strong antioxidant qualities, making them suitable for use in pharmaceutical products. In another study [31], the curcumin encapsulation system was considered to have a high potential for the transport and controlled release of this bioactive.

Between 2011 and 2024, a total of two ionic gelation-related patent applications were submitted to the National Institute of Industrial Property (INPI, Brazil): The process for creating alginate spheres is described in (1) Encapsulation process of oils and cosmetic products in alginate balls suspended in polysaccharide solution (2010); (2) Alginate spheres and methods of manufacture and use thereof (2019), which shows how different oils are encapsulated in alginate.

The literature lists spray drying as one of the encapsulation techniques for clove oil [32,33,34]. Nevertheless, the last approach presents difficulties for encapsulating essential oils due to their volatility and sensitivity to high temperatures, potentially leading to the loss of significant components [35]. Specific equipment is needed, which raises expenses [36]. The encapsulated emulsion’s deposition on the drying equipment’s wall is another limiting issue that raises costs and lowers yield and product quality [37]. Another popular technique is complex coacervation [38,39], however, it can be difficult to use on a large scale and can lead to variances in the final product, because it is easily agglomerated and offers little control over the size of the particles generated. In turn, encapsulation in liposomes necessitates the use of heating, which is problematic for the essential oil’s stability, in addition to tools and solvents that, depending on the concentration employed, may be hazardous [40].

High temperatures or strong organic solvents are not necessary with ionic gelation, a mild encapsulation technique that helps maintain the integrity and pharmacological activity of the encapsulated drugs. It does not require any sophisticated and/or expensive equipment. This method, which makes use of biodegradable and biocompatible polymers, is fast and simple to repeat. High encapsulation efficiency enhanced chemical stability, and regulated release of active substances are all possible with ionic gelation [3,4,41,42]. However, the disadvantages are that the generated encapsulated materials include a high proportion of water, which makes it easier for pathogenic microbes to contaminate them and lowers their thermal stability—two important aspects that lead to the product’s degeneration. The goal of using freeze-drying following ionic gelation is to address these problems [43]. High humidity also influences the release of encapsulated active ingredients, causing faster degradation of the protective polymer [44].

Therefore, we present a technological invention with new perspectives on the use of clove essential oil. The encapsulation process can reduce the limitations of the use of clove oil, which are related to factors such as high temperatures, low pressures, and exposure to air and light, among others, which could contribute to the decomposition or evaporation of active compounds. In addition to protecting against degradation under unfavourable environmental conditions, encapsulation can also extend the useful life of the essential oil [45]. Therefore, the objective of the present work was to develop sodium alginate encapsulates, filled with clove essential oil (CEO), using the ionic gelation technique, followed by freeze-drying for use as an antioxidant, protecting the CEO against oxidative stress and harmful environmental effects. Furthermore, we present promising data for future investigations of a new delivery system.

## 2. Results

### 2.1. Density and pH of CEO

CEO appears as a transparent, homogeneous, pale-yellow liquid, with an odor and flavor characteristic of eugenol, and a pH 5.2 (n = 3; ±0.3). The density value found in the present study, *d* = 1.11 g mL^−1^, was higher than that reported by other authors, 0.973 g mL^−1^ [46] and 0.989 g mL^−1^ [16]. Plant origin and growing conditions, post-harvest processing, and essential oil extraction methods can contribute to differences in density. Therefore, it is common to observe variations in the density of the same essential oil, especially if they are obtained from different sources and conditions. It is also important to highlight that the fraction rich in eugenol have a higher density. Isolated eugenol has a density of around 1.06 g mL^−1^, according to the National Toxicology Program (NTP), Institute of Environmental Health Sciences, National Institutes of Health.

### 2.2. Identification of Essential Oil Components

A significant chemical component of the CEO is eugenol. As indicated in Table 1, the presence of compounds like β-caryophyllene and humulene is especially noteworthy and may add to the essential oil’s complexity and efficacy [47].

Clove essential oil’s eugenol content varies based on several variables, including the extraction procedure, growing region, extraction method, and storage conditions. Generally speaking, at least 50% of the total concentration of clove essential oil should be made up of eugenol [47]. The CEO’s chromatogram can be seen in Appendix A of the Supporting Information.

### 2.3. Development of Encapsulates

The hardening of the particles occurs instantly, starting at the surface where the divalent cations react with the negatively charged biopolymer chains, forming a rigid three-dimensional structure, with a high-water content. Through this process, the ions diffuse into the particle, favouring cross-linking from the outside to the inside [48]. The resulting encapsulates were easily filled, spherical, and firm to the touch. The encapsulated ones were approximately 4.36 ± 0.30 mm (LF1), 4.30 ± 0.29 mm (LF2), 4.27 ± 0.30 mm (LF3). The development procedure was conducted in conditions that were suitable for achieving the intended shape. Owing to the oil-in-water emulsification process, the encapsulated materials generated with a larger proportion of CEO were likely opaque (Figure 1).

### 2.4. Freeze-Drying

To eliminate the aqueous component from the encapsulates and increase the final product’s thermal stability, freeze-drying was employed. The freeze-dried encapsulated products had become rigid and had a rough, wrinkled, and dry appearance (Figure 2). There was no clustering of the encased ones. The deformation is caused by the ice crystals that develop during freezing and then sublimate in the freeze dryer because of the decreased pressure, which causes mechanical stress on the particle [49].

### 2.5. Encapsulation Efficiency (EE%) and Moisture Content

The CEO’s encapsulation capacity was demonstrated by the lyophilized encapsulates. The highest percentage of encapsulation was observed for the LF3 (0.1%, *v*/*v*). High concentrations of essential oil have been shown to be poorly maintained in the alginate encapsulate. This finding might imply that the alginate polymer reaches its saturation point at low essential oil concentrations (Table 2). The encapsulates produced by ionic gelation had a moisture content of 89.00% + 0.02%.

### 2.6. Thermal Analysis (TGA/DSC)

The standard encapsulated (SE) (Figure 3A) thermogravimetric curve (TGA) shows three thermal stages related to gelled and freeze-dried sodium alginate. With a mass loss of 32.6%, the average temperature point in thermal stage I was found to be 110.4 °C. The start of the freeze-dried polymer’s deterioration may be connected to the mass loss in stage II (T = 231.2 °C; 14.7%). Following the three thermal stages, the sample’s mass loss was less than 57% (Figure 3A). Three heat absorption phases were found in the SE encapsulated contained in the calorimetric analysis: Δ*H* = +24.6 J g^−1^, Δ*H* = +15.2 J g^−1^, and Δ*H* = +11.2 J g^−1^ (Figure 3B).

Calorimetry (Figure 3B) LF3 demonstrated a discrete initial endothermic event, at 51.6 °C (Δ*H* = +1.43 J g^−1^). The average degradation temperature was 218.8 °C for encapsulated LF1, 224.0 °C for LF2, and 327.7 °C for LF3, and may be associated with the loss of essential oil. In pure essential oil, the average range of volatilization and/or decomposition occurs at a lower temperature, 179.2 °C. Detailed thermogravimetric and calorimetric data are represented in Appendix A.

### 2.7. Scanning Electron Microscopy (MEV)

Compared to those lyophilized without essential oil (Figure 4A–C) (LSE), the encapsulates lyophilized with essential oil at 1.0 (LF1) and 0.5% (LF2) (Figure 4D–I) developed more pleating and distinct cracks, which may indicate that the essential oil interfered with the degree of homogeneity of the formulation’s constituent parts. In comparison to the lyophilized material with a higher essential oil concentration, the 0.1% essential oil-containing material (Figure 4J–L) had a smoother surface and a more rounded appearance. The drying process is characterized by the presence of significant openings in the wall in image microscopy of the freeze-dried product without the oil.

The size of the particles analyzed ranged from approximately 2.7 mm ± 0.25 mm (LF1); 2.5 mm ± 0.21 mm (LF2); 2.5 mm ± 0.20 mm (LF3). This analysis is useful for classifying encapsulates at the macroparticle scale.

This encapsulation technique usually forms particles that vary in size, between 0.5 and 3.0 mm [18]. The size of the encapsulates formed is dependent on the diameter of the apparatus used, the viscosity and concentration of the biopolymeric solution, and the distance between the drip and the ionic solution [50].

### 2.8. FTIR

The alginate presented a broad band around 3300 cm^−1^, resulting from stretching vibrations of the -OH group [51]. The vibrations of the carbonyl of the alginate carboxylate group are responsible for the signals at 1596 and 1410 cm^−1^, which correspond to the symmetric and asymmetric stretching of the C=O carboxylate bond, respectively [52] (Figure 5). All the spectra of the freeze-dried encapsulates LF1, LF2, and LF3 showed alginate-specific signals. The spectra of the standard alginate (SE) samples (not shown) also displayed other prominent signals, like the bands located at 1125, 1085, and 1025 cm^−1^. These bands indicate the elongation of the C–O–C glycosidic bond found in the polymer chain [51,52,53,54].

Clove oil produced low-intensity signals generated from stretching vibrations of the O–H bond of the hydroxyl groups (3520–3433 cm^−1^). The bands resulting from the CH_3_ and CH_2_ were observed at low intensities (2975–2844 cm^−1^). Signals at 3071 and 3004 cm^−1^ indicate the stretching of a terminal alkene (H-C=) and aromatic ring in the composition of the C–H bond. These bands are found in alkenes like eugenol and other terpenes, like β-caryophyllene. As the functional groups of eugenol include double bonds, a strong band is shown in the range of 1502 cm^−1^ for C=C stretching. Eugenol showed a characteristic sharp peak at 1512 cm^−1^ attributed to the C=C stretching of the aromatic moiety that provides unique spectral information, with a significant contribution to the identification of the substance [55,56].

Double bond stretching signal (C=C) at 1639 cm^−1^, followed by, at 1512 cm^−1^, characteristic of an aromatic ring, are present in the spectra of LF1, LF2, and LF3. These groups are also confirmed by the vibrations that appear at 1462, 1452, and 1432 cm^−1^ and by the bands in the range between 995 and 818 cm^−1^, generated from folding involving C–H double bonds, including those adjacent to methine and methylene groups. The signals at 1266 and 1034 cm^−1^ of C–O stretching of hydroxyl groups and ethers, and other signals of folding of C–H bonds, are also present in all encapsulates. The bands presented in this spectrum were determined by consulting established references [55,57].

### 2.9. Degree of Swelling and Eugenol Release

The highest degree of swelling of the freeze-dried encapsulated (LF) was 19.2%, and occurred in saline phosphate buffer solution (10 mM, pH 7.4) after 30 min of immersion. After 60 min, this value was maintained and then the polymeric network began to fall apart, reducing the encapsulated mass. In the acidic solution (0.01 mol L^−1^ HCl with 2 mg mL^−1^ NaCl, pH 1.2), the degree of swelling was only 2.3% after 30 min and 4.7% after 60 min of immersion (Figure 6).

It is evident that in an acidic solution, the swelling process occurs, however, it occurs at a slower pace compared to the PBS solution, where the carboxylic groups tend to remain at the dissociated form, favoring ion exchange with the Ca^2+^ ions present in the encapsulated ones. Thus, there is a loss of integrity of the cross-links, increasing swelling. The presence of the phosphate ion and the high concentration of electrolytes in the PBS solution favor the ion exchange process and, consequently, the swelling of the encapsulated substances [58].

No macroscopic changes were noticed when the encapsulates were soaked in an acidic solution (Figure 7A). On the other hand, in PBS buffer solution, the encapsulates seemed to undergo slight dissolution almost immediately (10 min), making the solution opaque (Figure 7B).

Up to 25 min in saline phosphate buffer solution (10 mM, pH 7.4), the release of a higher percentage of eugenol occurred compared to the acidic solution (0.01 mol L^−1^ HCl with 2 mg mL^−1^ NaCl, pH 1.2); however, this percentage remains similar at the following times analyzed, increasing only after 240 min of analysis, with an 83.1% release at 300 min (Figure 8). On the other hand, in an acidic environment, the highest percentage of release occured after 25 min. Within 30 min, approximately 63.0% of the eugenol was released. The analysis ends in 240 min, releasing 92.5% eugenol. After this time, the eugenol values decreased. The encapsulated ones appeared to be more stable at low pH values, which may provide a promising pH-responsive release characteristic (Figure 8).

### 2.10. Determination of Total Phenol Content

The encapsulated LF1 presented 172.2 mg GAE g^−1^, a concentration considered sufficient to obtain important pharmacological activities (Table 3). In a 2018 study [59], CEO had a concentration of 9.07 mg GAE g^−1^ and obtained important in vitro antimicrobial activity against *Staphylococcus aureus*, *Escherichia coli*, *Listeria monocytogenes*, and *Salmonella typhimurium*, in addition to discrete antioxidant activity. Even in CEO samples with 0.18 mg GAE g^−1^, the DPPH radical scavenging activity was 25.2% [60]. At a concentration of 1.14 mg GAE g^−1^, CEO showed DPPH radical inhibition of 39.6% [61].

The encapsulated LF2 and LF3 (Table 3) showed no differences between them, but were statistically different from LF1 (*p* < 0.05).

### 2.11. Antioxidant Activity (DPPH)

The CEO demonstrated a strong ability to scavenge the DPPH radical (Steinhein, Germany), with an inhibition value reaching 87.4%, varying according to the concentration of the oil used. The concentration of essential oil capable of inhibiting the DPPH radical by 50% was 11.4 μg mL^−1^ (Table 4). The three formulations showed statistical differences in the concentrations analyzed by ANOVA and Tukey’s test (*p* < 0.05).

## 3. Discussion

The dripping angle, the diameter of the dripping hole, the viscosity of the polymeric solution, and the force of gravity are some of the factors that affect the size and shape of the encapsulate created by ionic gelation using sodium alginate [62]. The properties of the final product can also be influenced by the concentration and residence time of the compounds encapsulated in the calcium chloride solution [63]. We recommend that the encapsulated product be submerged in a calcium chloride solution for at least 24 h, to ensure effective chemical cross-linking between the sodium alginate and calcium chloride. This ionic bond creates a stable three-dimensional matrix, forming a gel capable of trapping the essential oil.

As noted by several authors, the produced encapsulates contain a high aqueous content (79.37 ± 0.06% [12]; 77.03 ± 0.01% [16]). Considering the significant water content remaining in the matrix (89% ± 0.02%), these figures are predicted. The encapsulated ones have less water in them and look smaller after freeze-drying. This low humidity lessens spoiling by preventing the growth of contaminating microorganisms in the formulations.

The procedure of freeze-drying the encapsulated preparations resulted in structural stiffness, which gave rise to a porous and irregularly shaped structure [17]. Despite this, the alginate preserved the active components. The porosity of the modified structures is most likely caused by the interactions between the cationic ions (Ca^2+^) and the anionic charge of the polysaccharide (COO-), which forms a three-dimensional gel network. Increased particle homogeneity, and a decrease in wall openings with the subsequent loss of active ingredients during storage, could be achieved by mixing in a surface-active agent like Tween 80.

The major constituents found in the essential oil were β-caryophyllene and eugenol (Table 1). These compounds have been linked to several pharmacological actions [16,17,18,46,47,48,49,50,55,57]. Other authors reported similar results [12,47,64,65,66,67,68,69,70]. These findings are reported in this study. The lyophilized encapsulates demonstrated the CEO’s strong encapsulating ability. The essential oil remains effectively trapped in the matrix, despite the decreased concentration. The EE% (encapsulation efficiency) of chitosan-encapsulated clove essential oil was only 39% in a recent study [71]. In the present work, the EE% for LF1, LF2, and LF3 in the current work was 39.3% ± 0.8, 50.4% ± 0.6, and 76.9% ± 0.5, respectively. Enhancing bioavailability and lowering dosages when used therapeutically are two benefits of high encapsulation rates, along with waste reduction and corresponding cost savings. Based on these data, it is suggested that the polymer matrix’s integrity may have been preserved through the freeze-drying process, leading to a better encapsulation efficiency. In terms of encapsulation, these obtained values show promise.

In thermal analysis, the initial step that corresponds to the standard encapsulation (SE) may be related to the beginning of coordinated molecular movements of the polymer chain, due to the increase in temperature, and the loss of residual moisture linked to the polysaccharide. Multiple –OH groups in the structure of sodium alginate promote hydrogen bonding with water. It is therefore possible for water from the surroundings to be adsorbed, even when the substance has been freeze-dried (Table 2). The sodium alginate matrix starts to break down when the temperature rises. The polymer’s sugar units break their bonds with one another, causing the structure to split. The polysaccharide’s glycosidic ring opens, releasing gases (CO_2_). Studies reveal that degradation starts at temperatures higher than 200 °C, which is a positive characteristic, showing high stability of the preparation [72,73,74,75].

The mass loss of the encapsulated SE after three thermogravimetric stages was less than 57% of the sample, which may suggest that the freeze-drying procedure reduced fast losses, resulting in the preservation of product integrity for an extended period. The first mass loss was noted at 50 °C in tests using freeze-dried alginate [76]. In contrast, this first loss in the current study began at 94.8 °C.

Second-order transitions in the calorimetric analysis (DSC) of the standard encapsulated product (LSE), that is, the one without CEO in the composition, (Figure 3B) arise from variations in the material’s heat capacity. The polymer chain’s relaxations during heat stress may be linked to the baseline shift. One explanation for the first calorimetric event of the encapsulated SE (Δ*H* = +24.6 J g^−1^), is that sodium alginate is releasing water. Water molecules evaporate when they acquire enough kinetic energy to overcome intermolecular force. The polymer degradation stages can involve the other endothermic bands of the standard composition. Several thermal transitions are occurring quickly after one another. As the freeze-dried substance breaks down, it can be exposed to heat.

Given that LF1’s initial mass loss was more intense than that of LF2 and LF3, this could be related to LF1’s higher moisture content. An endothermic event that may have been related to water loss was first shown by LF1 in Figure 3B. Three exothermic events were then observed, with the final one having the highest enthalpy energy (319.5 °C/Δ*H* = −35.6 J g^−1^). The asymmetric bands that have formed are most likely the result of variations in the sample’s heat capacity. The multiple early thermal breakdown events in LF2 (Figure 3B) produced strong exothermic energy.

The encapsulated LF3 showed a decreased mass loss, indicating that the low concentration of CEO in the product affected the formation of the emulsion that resulted in the encapsulated product, making it more thermally stable than the normal capsule. The polymer’s breakdown and the imprisoned CEO’s release could be connected to the exothermic heat release event.

Compared to the essential oil without polymeric protection, the freeze-dried encapsulates (LF) were effective in preserving the CEO, although oil loss occurred later. The LF1 freeze-dried product might be more practical because it had a higher essential oil content, which would be a competitive advantage, even though the LF3 freeze-dried product had a higher degradation temperature.

The presence of the CEO in the encapsulation was verified FTIR. The characterization of the encapsulates involved analyzing the chemical bonds and functional groups found in the constituent components, such as the oil and alginate. By analyzing spectroscopic signals generated by the vibrations of these bonds, it was feasible to ascertain the significant structural characteristics of these components. Xiao, Gu, and Tan [77] found that the inclusion of water in the alginate structure leads to signal saturation and increased broadening of the O–H band. The presence of alginate bands, which represent the mannuronic and guluronic acid blocks, may be observed in the spectral region at 780 to 1100 cm^−1^. This observation is consistent with the results reported by previous researchers [78,79,80].

Since the chromatographic analysis of the clove oil used in the formulations of encapsulated products revealed that two compounds, eugenol and β-caryophyllene, accounted for approximately 96% of its composition, it was expected that these compounds would predominate on the oil’s IR spectrum, with their vibrational signals determining the spectral profile.

The low intensity of some signals in the spectra obtained with encapsulated LF1, LF2, and LF3 did not allow for clear observation, primarily due to overlapping with the alginate -OH stretching band. However, the band at 2844 cm^−1^ in LF1 was observed to be faint. Nevertheless, additional signals emanating from the oil were detected in their spectra, as depicted in Figure 6. The enclosed spectra may exhibit slight differences, which can be attributed to overlaps, such as variations in intensities and band visualization. The characteristic bands of the CEO were observed in the encapsulated samples, without significant alterations, demonstrating that the essential oil was successfully incorporated into the polymeric matrix. This confirms the structural properties of the two main components of the oil (eugenol and β-caryophyllene) and indicates that they are well dispersed within the polymeric matrix.

Alginate is a polysaccharide obtained from seaweed. It has carboxylate groups (-COO-) in its structure, which bind to Ca^2+^ ions forming a gelatinous structure. The affinity of alginate for polyvalent cations is what determines the gelling property. Gelation is presented as an “egg crate” network. In this model, the polyvalent cations are linked coordinately with the carboxyl groups of guluronic acids. There is an electrostatic interaction between the cations of the crosslinking solution and the carboxylate anions of the G unit, while the rest of the alginate chain is maintained by van der Waals forces, resulting in a three-dimensional gel network. Therefore, the alginate concentration influences the mechanical properties of the gel formed [81,82].

In the present study, the encapsulated substances partially dissolved almost instantly upon contact with the PBS buffer solution, generating an opaque solution (Figure 8). Although the encapsulates did not swell in either the acidic or phosphate buffer media, the release of eugenol in both media was similar after 25 min. Initially, it can be seen that the release of eugenol in an acidic environment is low. This characteristic shows us that freeze-dried encapsulates can be useful in the controlled release of substances [83,84,85,86,87]. Drug release from alginate encapsulates is dependent upon the penetration of the dissolution medium into the spheres, the alginate matrix expanding and dissolving, and the active components dissolving via the enlarged matrix. The hydration and swelling that occur upon contact with the aqueous dissolution medium lead to the formation of a hydrogel surface layer, which controls the influx of the aqueous media and the dissolution of the active components [88]. Osmotic or mechanical pressures, enzymes, and/or pH variations are some of the stimuli that cause changes in the gel phase, responsible for the release of the active components that are contained in the alginate matrix [41]. The in vitro dissolution characteristics of apigenin-loaded beads based on calcium alginate were examined in a recent work [88]. The release of the drug was delayed with an increase in sodium alginate content. The apparent cross-linking points inside the granules increased when the alginate content was raised, resulting in a stiff matrix and postponing the release of the active components. Within a day, the spheres released the assets.

In fact, the charge of the particles encasing essential oils varies with pH levels [89]. The degree of ionization of the various functional groups of the carriers [90] changes as a result. For instance, the release of bioactive chemicals from chitosan nanocapsules loaded with peppermint, green tea [91], *Carum copticum* [89], and carvacrol [92] was substantially higher at pH 3.0 than it was at higher pH values (pH between 7 and 11) [90].

In the present work, compared to the PBS medium (pH 7.4), the initial release reported in the study happens somewhat later in an acidic medium (pH 1.2). The low stability of these capsules in a PBS medium serves as justification for this fact. About 58% of the eugenol in a PBS medium was released in 25 min. In contrast, this release was just 17% in an acidic medium. The release happened in a more regulated way and at comparable amounts in both media between 30 min and 2 h. In an acidic media, almost 83% had been released after 3 h, compared to roughly 63% in a PBS medium.

The microcapsules that were submerged in a pH 4.8 medium showed the highest rates of release with oregano essential oil. Under these circumstances, high release levels at predefined times (78% in 7 h) were noted. Lower release rates were observed in media with a pH of 4.2 (59% in 7 h). Intermediate rates were observed in media with a pH of 3.8 [93]. According to Muneratto, Gallo, and Nicoletti, the complex weakened, and release values increased at pH 4.8, where the zeta potential showed the smallest charge difference. The microcapsules were not destroyed though, so it can be suggested that the low initial release of eugenol in an acidic environment is related to the smaller difference in charge formed, thus making it more stable.

The degree of the swelling test was conducted to gain a deeper understanding of the behaviour of the capsules in acidic and PBS media. These data are crucial to optimize the permeability and stability of the encapsulation. When the encapsulated substances are placed in contact with the PBS buffer solution, the sodium ions present in the solution initiate an ion exchange process with the calcium ions linked to the -COO- groups of the MM units (β-D-mannuronic acid residues) of the alginate. Thus, the initial phase is the swelling of the encapsulates and the diffusion of calcium into the medium. As the immersion time increases, the encapsulates swell even more and the polymer chain becomes more relaxed, providing ion exchange with the more stable calcium ions, which are linked more strongly with the -COO- groups of the GG units (α-L-guluronic acid residues). When the swelling process reaches its maximum, the final phase is reached and the matrix collapses due to the loss of the “egg-box” structure caused by the disruption of the polymer cross-links and the release of calcium ions [58,94,95,96,97,98,99]. In the acidic environment, -COO- groups are converted to -COOH due to the low p*K*a of sodium alginate, about 3.2. Therefore, hydrogen bonds are easily formed between the -COOH groups, which makes it difficult for water molecules to penetrate the alginate–calcium matrix. This explains the low swelling of the encapsulates in an acidic environment [97].

The values of total phenolic compounds described in the literature for CEO are not homogeneous: 898.9 mg GAE g^−1^ [100], 845 mg GAE g^−1^ [101], and 9.07 mg GAE g^−1^ [59]. Clove extracts have a high phenolic content, with 230 mg GAE g^−1^ extract, and show antioxidant properties ranging from 25.3 to 91.4% [102]. These changes are to be expected, given that the essential oil’s chemical content varies depending on the plant component employed, the extraction technique, and the sample’s overall properties [103]. In comparison to the other encapsulated products, the LF1 encapsulated had a superior phenolic concentration, ensuring a sufficient concentration to yield significant pharmacological activity.

The literature reported varying concentrations of essential oil (ranging from 10.9 to 380 μg mL^−1^) that might block the DPPH radical by 50% [16,100,101,102,103,104,105]. These numbers cover the essential oil’s IC_50_, LF1, and LF2, as found in the current investigation. It has been shown by other authors [106] that 484.7 μg mL^−1^ of clove essential oil was required to prevent 94.9% of DPPH radiation. These findings imply that there can be wide variations in the pharmacological activity of products containing essential oils. Because most synthetic antioxidants are hazardous and carcinogenic in small amounts, they are unable to shield the body from free radical attacks [107,108]. Consequently, using natural antioxidant chemicals might aid in the removal of free radicals.

In some studies [109], the antioxidant activity of clove extract was shown to be more than 10 times greater than that of vitamin E in the DPPH free radical scavenging capacity test. The strong antioxidant capacity of CEO can be attributed to the hydrogen donating capacity, presented by a wide range of available chemical constituents, in the essential oil, especially phenols and terpenoids, such as eugenol and β-caryophyllene.

β-caryophyllene demonstrates strong antioxidant activity, breaking chains and eliminating free radicals, such as hydroxyl and superoxide anion radical, which are highly reactive [110,111,112]. The literature describes eugenol’s ability to inhibit the production of superoxide anion in neutrophils, induced by fMLF (factor N-formyl-methionyl-leucyl-phenylalanine) or PMA (phorbol myristate acetate), blocking the phosphorylation of the cell signaling pathway involved in the production of ROS (reactive oxygen species), Raf/MEK/ERK1/2/p47phox. When activated by stimuli such as fMLF, this signaling pathway leads to the phosphorylation of key proteins such as Raf, MEK, ERK1/2, and p47phox, which results in the activation of the NADPH oxidase complex and consequent production of superoxide radical anion [113,114]. Furthermore, eugenol can increase the activity of antioxidant enzymes, such as SOD-3 or GST-4, reducing the accumulation of ROS in vivo and exerting antioxidant effects [115,116]. Additionally, its Fe^3+^ reducing properties and electron-donating capacity can neutralize free radicals, forming stable products to reinforce its antioxidant effects [104,116,117].

Eugenol exhibits a greater inhibitory effect on hydrogen peroxide than other ROS and can also block DNA oxidation and hydroxyl radical-induced lipid peroxidation [118]. These properties highlight eugenol as a potentially valuable agent in protecting against cellular oxidative damage. Eugenol increases the transcriptional activity and expression level of nuclear factor erythroid 2-related factor 2 (Nrf2), a central regulator of cellular responses to oxidative stress, in a dose-dependent manner. Additional studies revealed that eugenol improved Nrf2 stabilization and nuclear translocation. Eugenol treatment reduces intracellular ROS levels, while increasing cellular resistance to H_2_O_2_ in an Nrf2-dependent manner [119].

The antioxidant activity of the essential oil can be preserved if the encapsulation process is carried out properly, avoiding degradation by oxygen and temperature [120]. CEO is encapsulated in a sodium alginate-containing polymeric framework, which shields the phenolic components and may lessen degradation reactions that are aided by contact with the external environment. The three formulations of CEO maintained their antioxidant power, however, LF1 showed findings that were almost identical to those of pure essential oil, with an IC_50_ value of 18.1 μg mL^−1^, reaching 80.3% inhibition. When comparing the freeze-dried data to clove oils reported in the literature, the former demonstrated higher antioxidant effectiveness. The study’s findings indicate that lyophilized CEO encapsulates exhibit a noteworthy reducing power, suggesting a promising antioxidant capacity with the potential to yield moderate to high benefits when used as a natural antioxidant.

CEO-loaded alginate freeze-dried encapsulates may be promising against oxidative stress. Furthermore, alginate works as an efficient asset protection system and can be useful in controlling the release of encapsulated content. The literature [121,122] shows us that alginate is a non-toxic, biocompatible, and biodegradable natural polymer, being considered a potential vehicle for encapsulating compounds. This work presents an innovative product that is rich in eugenol, easy to reproduce, safe for oral consumption in pre-established concentrations [123], and thermally stable, which can increase the useful life of imprisoned compounds [124], with characteristics favorable to its use.

## 4. Materials and Methods

### 4.1. Materials

Sodium alginate and pharmaceutical-grade calcium chloride for food and pharmaceutical use were obtained from the Alquimia pharmacy (Indaiatuba, Brazil). The acidic medium (0.01 mol L^−1^ hydrochloric acid with 2 mg mL^−1^ of NaCl, pH 1.2) was obtained from Exodo Científica (Sumaré, Brazil) and the saline phosphate buffer solution (10 mM, pH 7.4) was obtained from Sigma-Aldrich Brazil, Ltd. (São Paulo, Brazil). Clove essential oil was extracted from the dried leaves and floral buds of the plant using the steam drag technique, obtained commercially from the company Quinarí (Ponta Grossa, Brazil). Clove essential oil comes from Indonesia (Registration with ANVISA: 25351.183090/2017-50; CAS number: 8000-34-8; 84961-50-2/Chemical Abstracts Service).

### 4.2. Density and pH of CEO

The density of CEO was determined using a pycnometer with a nominal volume of 10 mL (20 °C), previously calibrated with distilled water. The pH analysis was carried out using a digital pH meter (Quimis^®^ Q400MT, Diadema, Brazil), using a glass electrode previously calibrated in buffer solutions of pH 7.0 and 4.0.

### 4.3. Identification of Essential Oil Components

The identification of the components present in the essential oil was carried out on a Shimadzu GCMS-QP 2010 Plus Chromatograph at the Medicines Technology and Control Laboratory (UFAL/Alagoas/Brazil). The stock solution was prepared in hexane (10, 25, 50, 75, and 100 μg mL^−1^), using nonpolar silica chromatographic capillary column, with helium as the carrier gas (1 mL min^−1^). The oven temperature was programmed as follows: 60 °C maintained for 1 min, then 3 °C/min until 90 °C, and finally, 30 °C/min at 210 °C with 15 min retention time, using a spitless injection mode (1 µL). The injector and the transfer line temperatures were 250 and 280 °C, respectively. The source and quadrupole temperatures were set at 230 and 150 °C. Furthermore, the mass spectra were obtained at an ionization voltage of 70 eV and the mass range was *m*/*z* 40–400 u.

### 4.4. Development of Encapsulates

Clove essential oil (CEO) was encapsulated using the ionic gelation technique with 1.0% (*w*/*v*) sodium alginate. The aqueous solutions were left under stirring for 30 min at 1200 rpm in an IKA RW20 digital mechanical propeller stirrer. Essential oil (0.1%, 0.5%, and 1.0% *v*/*v*, relative to the alginate solution) was added to the alginate solutions and emulsified (3 min, 1200 rpm). The formed emulsion was dropped with a Pasteur pipette onto the calcium solution (3.0%, *w*/*v*), under constant stirring (15 min, 200 rpm), at a temperature of 25 °C. The encapsulated capsules formed were removed from the solution with the aid of tweezers, then named F1 (1.0% CEO), F2 (0.5% CEO), and F3 (0.1% CEO), and stored in a freezer for subsequent freeze-drying. The size of the encapsulates formed was obtained by the image processing and analysis Software ImageJ (version 1.54i).

### 4.5. Freeze Drying

The encapsulated samples were frozen in a −20 °C freezer for 24 h and taken to the Terroni LS 3000 freeze-dryer, under conditions of pressure 5 mmHg and temperature of −50 °C, for another 24 h. The dehydrated encapsulates were stored in a desiccator and were named encapsulated LF1 (1.0% CEO), LF2 (0.5% CEO), and LF3 (0.1% CEO).

### 4.6. Encapsulation Efficiency

To determine the amount of CEO trapped in the polymeric matrix, 0.1 g of the encapsulated compounds were dissolved in 2.0 mL of distilled water, with the aid of ultrasound (15 min). After extraction in hexane, the samples were filtered and injected into GC/MS for analysis under the conditions described previously (Section 4.3). The amount of encapsulated essential oil and the amount lost in the calcium chloride solution were calculated:(1)$$EE\%=\frac{NE}{eCEO}\times 100,$$
where EE = Encapsulation efficiency; NE = non-encapsulated eugenol; eCEO = eugenol in clove essential oil.
(2)$$OS\%=\frac{ES}{EO}\times 100,$$
where OS = Essential oil in the supernatant; ES = eugenol content in the supernatant; EO = eugenol content in essential oil.

### 4.7. Moisture Content

The encapsulated samples (1.0 g) were subjected in triplicate to infrared in a Shimadzu model MOC63u Moisture Analyzer (Tokyo, Japan).

### 4.8. Thermal Analysis

The analyses were obtained using a Shimadzu model TGA-50H and DSC-60. The temperatures were from 25 to 500 °C or 900 °C (10 °C/min), under a nitrogen atmosphere and flow rate of 20 mL/min or 50 mL/min, using 5.0 ± 0.5 mg of samples. The data were analyzed using Shimadzu’s Tasy software (version 2.21).

### 4.9. MEV

The study of the morphology of the freeze-dried encapsulates was carried out using a TESCAN VEGA3 microscope at the Federal Institute of Alagoas (Maceió, Brazil), with an accelerated voltage of up to 15 kV. The encapsulated samples were fixed with double-sided carbon adhesive metallic tape on a circular steel stub, then metallized on a gold target for 200 s (45 mA current) to obtain the images. The scan was carried out at different points with variable focus magnification, with the aim of observing the shape, porosity, and the presence of cracks and deformations.

### 4.10. FT-IR Characterization

The samples were characterized by Fourier Transform Infrared Spectroscopy (FT-IR) on a Agilent Cary 660 Spectrometer (Santa Clara, CA, USA), equipped with an attenuated total transmittance (ATR) module with zinc selenide (ZnSe) crystal. A range of 4000–650 cm^−1^ was used in 48 scans. The materials (encapsulated, CEO, and alginate) were added to the ATR crystal and the spectra were acquired with the Resolution Pro software 5.4.1. The data obtained were compiled in the Origin^®^ Software version 8.0.

### 4.11. Degree of Swelling and Eugenol Release

The swelling capacity of the encapsulated substances was evaluated by the percentage of weight variation when they were subjected to an acidic medium (0.01 mol L^−1^ hydrochloric acid with 2 mg mL^−1^ NaCl, pH 1.2), and saline phosphate buffer solution (10 mM, pH 7.4). A total of 0.1 g of the lyophilized encapsulates and 10 mL of the solutions were used; they were kept at room temperature (25 °C) and low agitation (1200 rpm). At time intervals of 10, 25, 30, and 60 min, the encapsulates were removed with the aid of tweezers, dried on paper towels, and weighed on an analytical balance. To calculate the degree of swelling, the weight variation of the samples before and after immersion in the medium was considered (Equation (3)).
(3)$$Degree\;of\;swelling=\frac{\left(Pf-Pi\right)}{\left(Pi\right)}\times 100,$$
where Pf = final weight of the encapsulated products, Pi = initial weight of the encapsulated products.

To analyze the percentage release of essential oil retained in the lyophilized encapsulates, the acid and PBS media were collected at different times (10, 25, 30, 60, 120, 180, 240, and 300 min) and subjected to extraction in hexane (proportion 1:1). The analyzes were carried out in GC/MS, under the same conditions described in Section 4.2. The graphics were obtained using Origin^®^ Software version 8.0.

### 4.12. Determination of Total Phenol Content

The determination of total phenol content was carried out in triplicate, according to the Folin–Ciocaulteu method [125], using gallic acid (Sigma Aldrich, Berlin, Germany) as standard. The reaction took place in a water bath at 50 °C (20 min) and was read on a UV-Vis spectrophotometer, model 1240 Shimadzu, λ = 760 nm. Samples were analyzed at a concentration of 25 μg mL^−1^.

### 4.13. Antioxidant Activity (DPPH)

The antioxidant activity was determined by the DPPH method [126]. A 0.1 mM solution of the DPPH radical was prepared in absolute ethanol and stored in amber glass. In 5.0 mL flasks, 2.0 mL of the DPPH solution was added and at 1-min intervals for each flask, an aliquot of the samples, containing 5, 10, 25, 30 and 40 μg mL^−1,^. The volume was completed with absolute ethanol. The reaction remained in the dark for 30 min, and was read on a UV-Vis spectrophotometer, λ = 517 nm.

The percentage of antioxidant activity was calculated based on the following equation:(4)$$\%DPPH•remaining=\frac{\left(Abs\;sample-Abs\;solvent\right)}{\left(Abs\;control-Abs\;solvent\right)}\times 100,$$
where Abs sample = absorbance of the reaction between the DPPH radical solution and the sample to be analyzed; Abs solvent = absorbance of the solvent solution used to prepare the sample to be analyzed; Abs control = absorbance of the DPPH radical with an aliquot of the solvent, corresponding to the volume of the highest concentration of the sample.

After determining the remaining DPPH radical, the percentage of inhibition of the DPPH radical was determined using the following formula:(5)%DPPH radical inhibition=100−%DPPH•remaining

## 5. Conclusions

With a low concentration of sodium alginate (1.0%), the ionic gelation process created a three-dimensional network that effectively imprisoned the CEO. The high water content was eliminated, the encapsulates were given thermal stability, and a high percentage concentration of active ingredients was obtained through the application of the freeze-drying procedure. The LF3 encapsulate was the most resistant to the rise in temperature, however, all of the CEO concentrations utilized in the encapsulates showed thermal stability. The LF3 that was encased demonstrated the maximum encapsulation efficiency. The eugenol spectroscopic bands were detected in the polymeric matrix, with excellent dispersion and without any molecular alterations. No significant apertures in the encapsulating wall were found, displaying the wrinkling typical of freeze-drying.

The encapsulated essential oils retained their antioxidant potential, with LF1 exhibiting the highest phenolic concentration and being the most effective. The encapsulated substance exhibited the ability to release its active components in a phosphate buffer solution, as well as in acidic media. When beginning in vivo research, these facts can be considered. Finally, this work demonstrates the ionic gelation technique with subsequent freeze-drying as a feasible, quick, and inexpensive method for the effective encapsulation of volatiles, which can be an alternative for the preservation of essential oils of products. We now understand the challenges associated with the conservation and storage of essential oils. Positive test results made room for additional investigation of this substance.

Clove encapsulates have a strong antioxidant profile, which makes them suitable for use as dietary supplements, or in food packaging. They can also help prolong the shelf life of food goods by shielding them from oxidation and degradation. These clove encapsulates have additional uses in cosmetics, where they can shield the skin from free radical damage. They can also be utilized in medicine to treat oxidative stress-related illnesses, like neurological and cardiovascular disorders. To determine the right dosage for each purpose, and compatibility with other ingredients in various formulations, more research is necessary.

## Figures and Tables

**Figure 1 pharmaceuticals-17-00599-f001:**
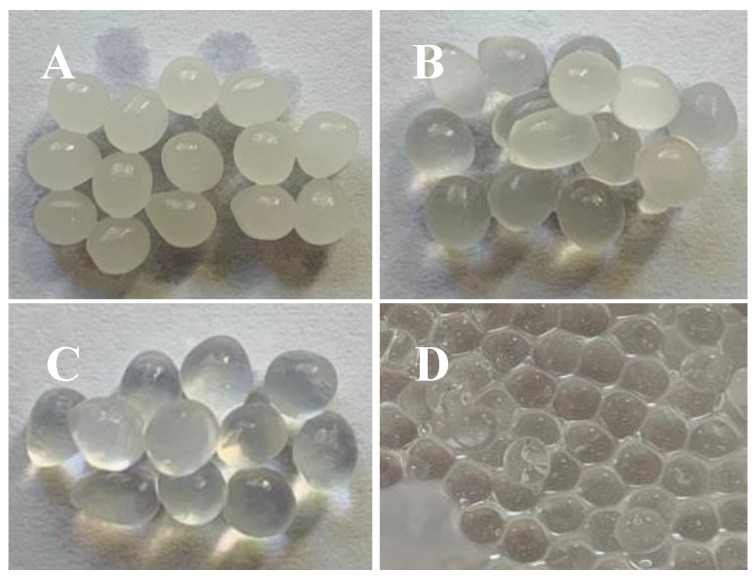
Encapsulated products obtained by ionic gelation. (**A**) F1 (1.0% essential oil); (**B**) F2 (0.5% essential oil); (**C**) F3 (0.1% essential oil); and (**D**) sodium alginate without essential oil (standard encapsulated − SE). The encapsulated ones were approximately 4.36 ± 0.30 mm (LF1), 4.30 ± 0.29 mm (LF2), and 4.27 ± 0.30 mm (LF3).

**Figure 2 pharmaceuticals-17-00599-f002:**
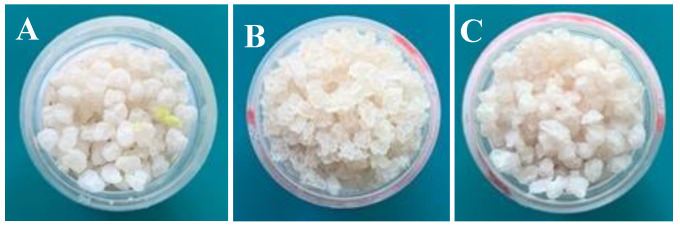
Encapsulates after freeze-drying. (**A**) LF1 (1.0% essential oil); (**B**) LF2 (0.5% essential oil); (**C**) LF3 (0.1% essential oil). Particle size: 2.7 mm ± 0.25 mm (LF1); 2.5 mm ± 0.21 mm (LF2); 2.5 mm ± 0.20 mm (LF3).

**Figure 3 pharmaceuticals-17-00599-f003:**
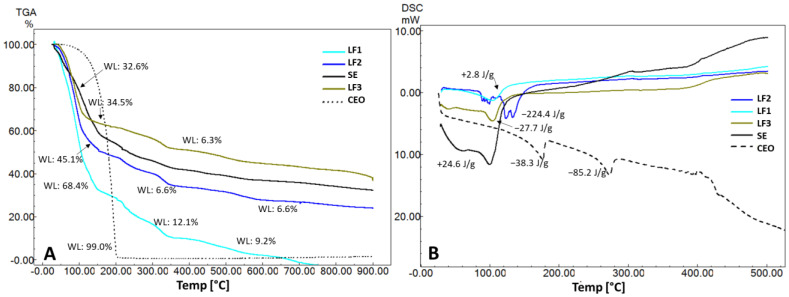
The thermogravimetric (**A**) and calorimetric (**B**) curves show, respectively, the mass loss (%) or the energy variation of each sample as a function of temperature (°C). SE: standard encapsulated.

**Figure 4 pharmaceuticals-17-00599-f004:**
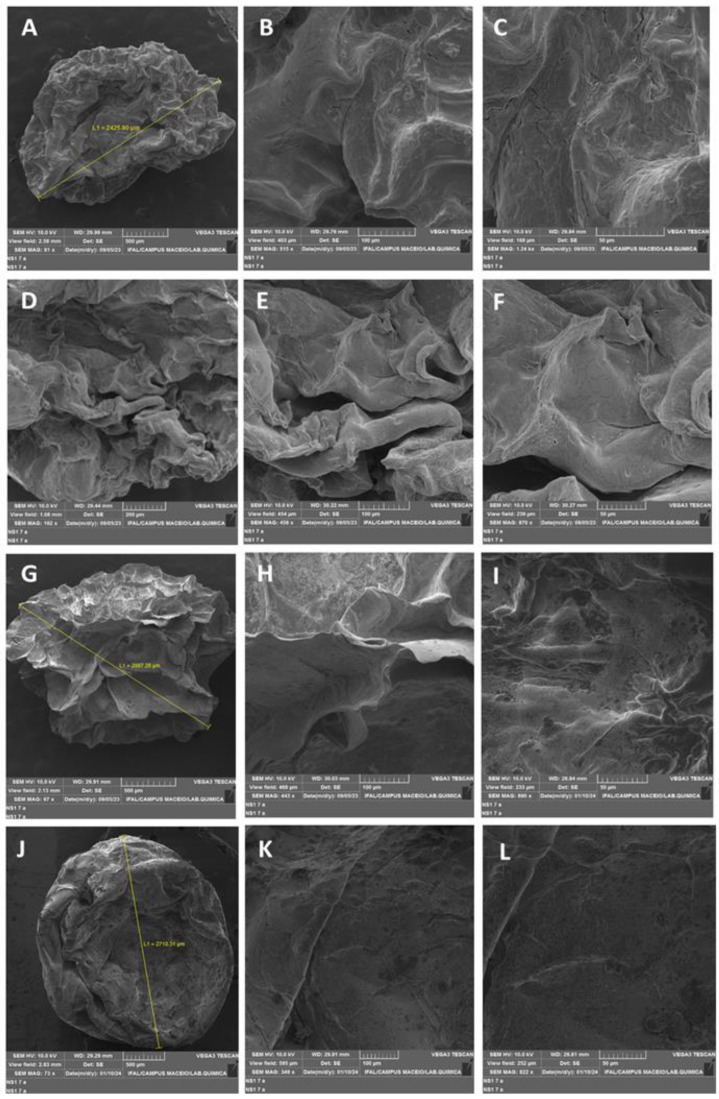
Scanning electron microscopy of freeze-dried samples at different points with variable focus magnification. (**A**–**C**) Standard freeze-dried encapsulated (LSE), (**A**) view field—2.58 mm, 81x, WD—29.99 mm; (**B**) view field—403 µm, 515x, WD—29.79 mm; (**C**) view field—168 µm, 1.24 kx, WD—29.94 mm. (**D**–**F**) encapsulated LF1 (1.0%), (**D**) view field—1.08 mm, 192x, WD—29.44 mm; (**E**) view field—454 µm, 458x, WD—30.22 mm; (**F**) view field—239 µm, 870x, WD—30.27 mm. (**G**–**I**) encapsulated LF2 (0.5%), (**G**) view field—2.13 mm, 97x, WD—29.91 mm; (**H**) view field—468 µm, 443x, WD—30.03 mm; (**I**) view field—233 µm, 890x, WD—28.84 mm. (**J**–**L**) encapsulated LF3 (0.1%), (**J**) view field—2.83 mm, 73x, WD—29.29 mm; (**K**) view field—595 µm, 349x, WD—29.81 mm; (**L**) view field—252 µm, 822x, WD—29.81 mm.

**Figure 5 pharmaceuticals-17-00599-f005:**
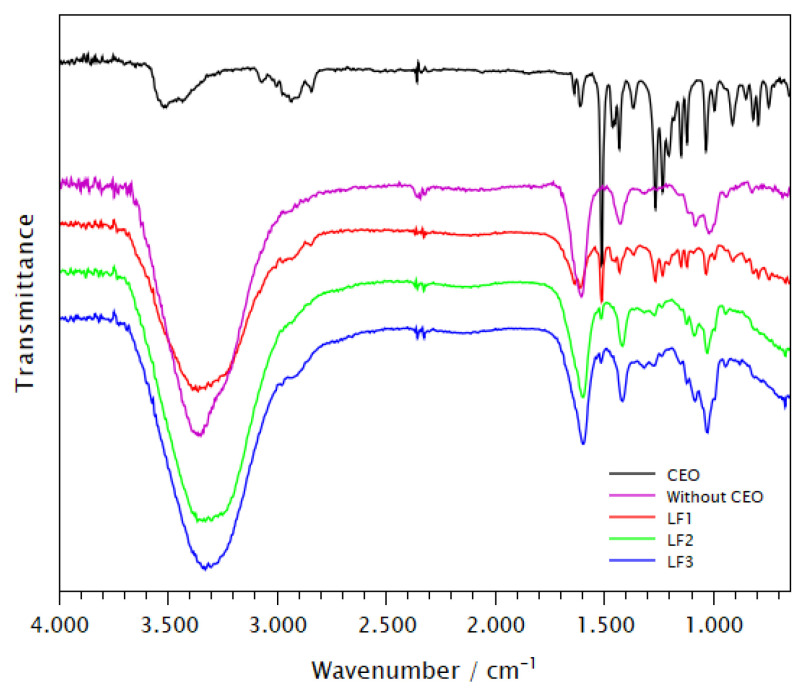
The infrared spectra of CEO, and lyophilized encapsulates LF1, LF2 and LF3 were obtained by attenuated total transmittance (ATR).

**Figure 6 pharmaceuticals-17-00599-f006:**
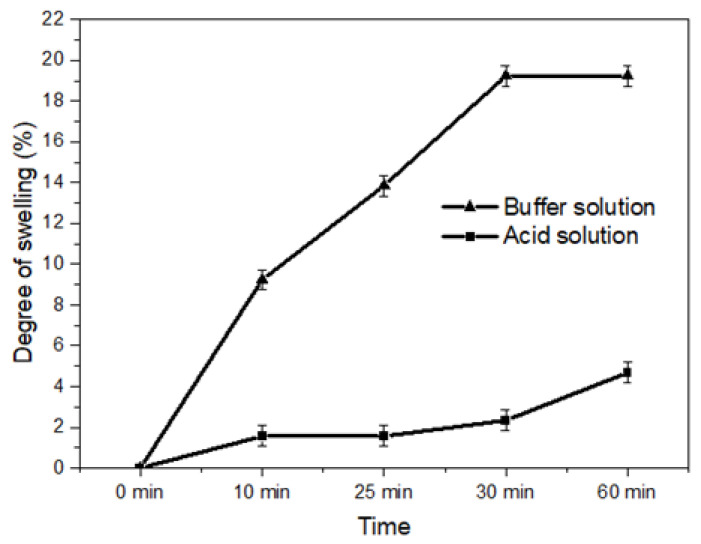
Degree of swelling of LF1 at different immersion times in acid solution and phosphate buffer**.** To calculate the degree of swelling, the weight variation of the samples before and after immersion in the media was considered.

**Figure 7 pharmaceuticals-17-00599-f007:**
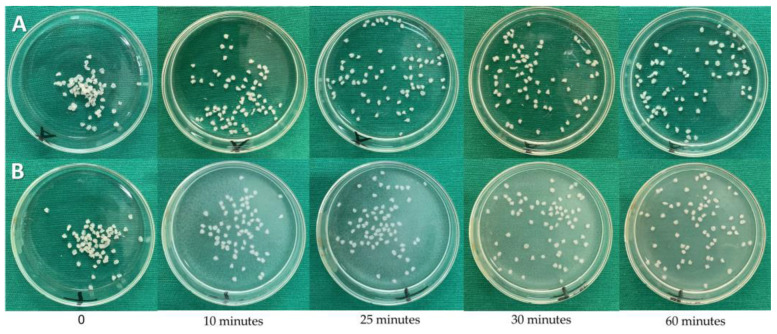
Image of LF1 encapsulated in acid solution (**A**) and buffer solution (**B**) at different immersion times. Eugenol release was measured through successive GC/MS analyzes (conditions described in Section 4.11).

**Figure 8 pharmaceuticals-17-00599-f008:**
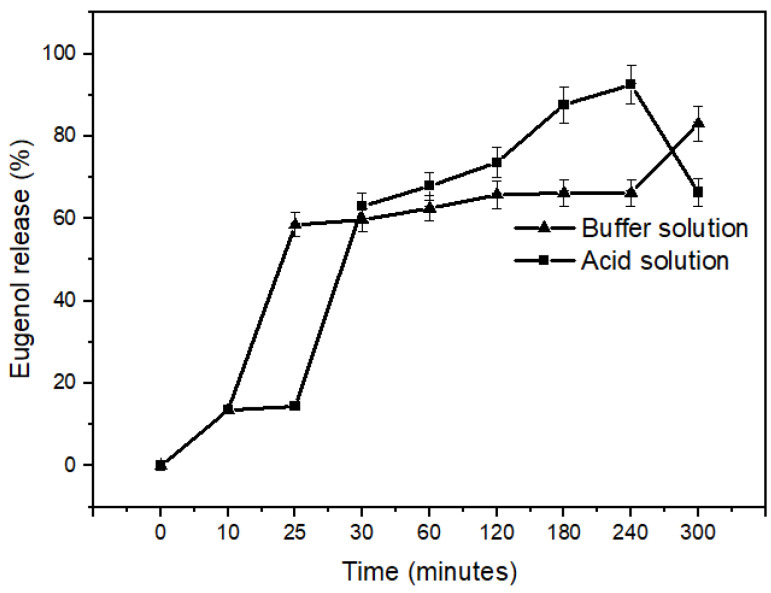
Release of eugenol from encapsulated LF1 in acid solution and phosphate buffer at different times. Data were obtained using GC/MS under the conditions described in Section 4.11.

**Table 1 pharmaceuticals-17-00599-t001:** Identification of the main chemical components of clove essential oil (*vide* 4.3).

Peak	RT	Area	Component	Chemical Structure	Content (%) *
1	10.28	79,065,820	Eugenol	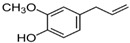	83.7
2	10.87	6,521,051	Caryophyllene	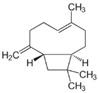	13.8
3	11.17	1,153,276	Humulene	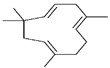	2.5

RT: retention time. * Corresponds to the relative area of the peak.

**Table 2 pharmaceuticals-17-00599-t002:** Encapsulation efficiency (EE%) of clove essential oil (CEO).

	LF1	LF2	LF3
% incorporated	39.3 ± 0.8 *	50.4 ± 0.6 *	76.9 ± 0.5 *
Moisture content	4.7 ± 0.05%	4.1 ± 0.09 *	3.9 ± 0.05 *

LF1: 1.0% (*v*/*v*) CEO; LF2: 0.5% (*v*/*v*) CEO; LF3: 0.1% (*v*/*v*) CEO. * ±SD: standard deviation.

**Table 3 pharmaceuticals-17-00599-t003:** The total phenol content identified in freeze-dried encapsulates was obtained under the conditions described in Section 4.12.

	Total Phenols (mg GAE g^−1^) *
Essential oil	449.9 ± 0.08 ^a^
LF1	172.2 ± 3.85 ^b^
LF2	147.6 ± 2.86 ^c^
LF3	146.5 ± 5.54 ^c^

* average + coefficient of variation. GAE = gallic acid equivalent (m/m). Different letters indicate statistical differences (*p* < 0.05).

**Table 4 pharmaceuticals-17-00599-t004:** DPPH antioxidant activity of clove essential oil and freeze-dried encapsulates (% DPPH radical inhibition).

Concentration (µg mL^−1^)	(% DPPH Inhibition) *^,1^			
	CEO	LF1	LF2	LF3
5	31.9 ± 1.0	28.2 ± 0.2	11.2 ± 1.9	5.8 ± 0.6
10	50.8 ± 0.3	45.4 ± 1.3	24.8 ± 1.4	14.1 ± 1.6
25	78.5 ± 1.0	69.2 ± 0.8	48.8 ± 0.6	20.2 ± 1.6
30	82.9 ± 0.8	75.9 ± 0.5	54.2 ± 0.2	22.3 ± 0.4
40	87.4 ± 0.3	80.3 ± 0.6	65.1 ± 0.7	30.8 ± 0.3
IC_50_ (µg mL^−1^)	11.4 ^a^	18.1 ^b^	25.2 ^c^	86.4 ^d^

* Average ± Coefficient of variation. IC_50_ = ability to scavenge the DPPH radical by 50%. Different letters indicate statistical differences (*p* < 0.05). ^1^ For this analysis, a 0.1 mM DPPH radical solution was prepared in absolute ethanol.

## Data Availability

Data is contained within the article and Appendix A.

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
