# Peer review of "Insights from Syzygium aromaticum Essential Oil: Encapsulation, Characterization, and Antioxidant Activity"

_pharmaceuticals, 2024, doi:10.3390/ph17050599_

Round 1

Reviewer 1 Report

Comments and Suggestions for Authors

First of all, congratulations for the good work done. It is a very complete work and it is very easy to understand. I just wanted to make some comments and if it is possible to answer some questions. Thanks in advance.

1. In section 4.1 Materials, it would be convenient to include the components of the PBS and the acid medium used for the swelling and release test.

2. I believe that Table 6 does not provide very relevant information, so it could be eliminated since part of the information contained in it is also included in section 2.1.

3. In the title 4.2 and 2.1 remove the comma after the word “Density”.

4. In section 4.4. It is not stated at what temperature the ionic gelation process was carried out. How were the formed encapsulates separated or collected?

5. In section 4.5., in the freeze-drying process, only primary desiccation was carried out?

6. In section 4.11, as described, it appears that the essential oil release test was only carried out in the acid medium; it should be specified that it was also carried out with PBS.

7. I consider that subsection 4.13.1 is not necessary, and it is sufficient to include only section 4.13 Antioxidant activity (DPPH). The same occurs in section 2.11.1.

8. In formula 4 the first parenthesis of the numerator is missing.

9. I consider that Table 2 is unnecessary, since its content is explained in material and methods and the only difference between the different types of samples is the % of CEO.

10. Have the encapsulation sizes for each of the conditions not been determined independently? There could be differences in sizes when changing the CEO %.

11. In table 3, the relationship between % incorporated and % not incorporated is not well understood, since there should be a material balance. If it's not included in encapsulates, it should be out.

12. Above the caption of figure 8 there is a mistake and a number 300 appears.

13. On line 487 there is a missing period at the end of the sentence.

14. In Table S1, decimals appear with commas instead of periods.

15. They must adapt the list of references to the format of the journal. Separation of authors by semicolon, in all cases abbreviation of the journal, separation of year and volume by comma...

Author Response

First of all, I would like to thank all the reviewers for their careful analysis that allowed me to improve the present article

  • Author's Reply to the Review Report (Reviewer 1)

First of all, congratulations for the good work done. It is a very complete work, and it is very easy to understand. I just wanted to make some comments and if it is possible to answer some questions. Thanks in advance.

  1. In section 4.1 Materials, it would be convenient to include the components of the PBS and the acid medium used for the swelling and release test.

Answer: We added the PBS and acid data in Section 4.1.

  1. I believe that Table 6 does not provide very relevant information, so it could be eliminated since part of the information contained in it is also included in Section 2.1.

Answer: We removed Table 6. Thanks

  1. In the title 4.2 and 2.1 remove the comma after the word “Density”.

Answer: We removed the comma after the words “density” in headings 4.2 and 2.1.

  1. In section 4.4, it is not stated at what temperature the ionic gelation process was carried out. How were the formed encapsulates separated or collected?

Answer: We add the temperature used in the ionic gelation process. We included how the encapsulates were collected.

“The formed emulsion was dropped with a Pasteur pipette onto the calcium solution (3.0%, w/v), under constant stirring (15 minutes; 200 rpm), at a temperature of 25°C. The prepared capsules were removed from the solution with the aid of tweezers, named F1 (1.0% CEO), F2 (0.5% CEO), and F3 (0.1% CEO), and stored in a freezer for subsequent freeze-drying”.

  1. In section 4.5., in the freeze-drying process, only primary desiccation was carried out?

Answer: Yes, we only use primary desiccation, where there is freezing and then sublimation, removing the ice under vacuum. As a result, the residual moisture was 4.7% (LF1), 4.1% (LF2), and 3.9% (LF3).

  1. In section 4.11, as described, it appears that the essential oil release test was only carried out in the acid medium; it should be specified that it was also carried out with PBS.

Answer: We corrected Section 4.11. We included the fact that PBS was also used for the release assay.

  1. I consider that subsection 4.13.1 is not necessary, and it is sufficient to include only section 4.13 Antioxidant activity (DPPH). The same occurs in section 2.11.1.

Answer: Thanks. We removed subsection 4.13.1, leaving only “4.13”. The same correction was made in subsection 2.11.1.

  1. In formula 4 the first parenthesis of the numerator is missing.

Answer: We add the parenthesis.

  1. I consider that Table 2 is unnecessary, since its content is explained in material and methods and the only difference between the different types of samples is the % of CEO.

Answer: We removed table 2.

  1. Have the encapsulation sizes for each of the conditions not been determined independently? There could be differences in sizes when changing the CEO %.

Answer: We add the particle size for each sample, in each condition.

  1. In table 3, the relationship between % incorporated and % not incorporated is not well understood, since there should be a material balance. If it's not included in encapsulates, it should be out.

Answer: As suggested, we removed the CEO's unincorporated %. This percentage refers to the amount of oil in the calcium chloride solution, where the encapsulates were formed.

  1. Above the caption of figure 8 there is a mistake and a number 300 appears.

Answer: We did not identify the error in number 300. We previously had the phrase “However, this percentage remains similar at the following times analyzed, increasing only after 240 minutes of analysis, with 83.1% release at 300 minutes (Figure 8)”.

  1. On line 487 there is a missing period at the end of the sentence.

Answer: We add the full stop to the end of line 487.

  1. In Table S1, decimals appear with commas instead of periods.

Answer: We replaced commas with periods in Table S1.

  1. They must adapt the list of references to the format of the journal. Separation of authors by semicolon, in all cases abbreviation of the journal, separation of year and volume by comma...

Answer: References were adapted according to the magazine.

Thanks again.

Reviewer 2 Report

Comments and Suggestions for Authors

This study describes the encapsulation of clove essential oils in alginate balls and their characterization as antioxidants.  While this paper contributes to advancing pharmaceutical technologies, I have some reservations that should be addressed before publication.  My comments are as follows:

1.     The macrocapsules prepared by the reaction of sodium alginate with calcium chloride are broadly used in the food and pharmaceutical industries.  The authors should discuss the significance of these macrocapsules and their industrial applications, comparing them to other microcapsules in the Introduction.

2.     The release of the essential oils from the capsules should be dependent on pH.  I recommend that the authors investigate the pH-dependent release.

Author Response

  • Author's Reply to the Review Report (Reviewer 2)
  • Thanks a lot for the careful analysis.

This study describes the encapsulation of clove essential oils in alginate balls and their characterization as antioxidants.  While this paper contributes to advancing pharmaceutical technologies, I have some reservations that should be addressed before publication.  My comments are as follows:

  1. The microcapsules prepared by the reaction of sodium alginate with calcium chloride are broadly used in the food and pharmaceutical industries.  The authors should discuss the significance of these microcapsules and their industrial applications, comparing them to other microcapsules in the Introduction.

Answer: We added this information in the fifth paragraph of the introduction:

"Ionic gelation encapsulates of alginate are naturally biodegradable, perhaps more sustainable, and less expensive than other forms of encapsulates reported in the literature, such as those based on synthetic polymers [19,20]. Alginates are highly common polymers in the food business; they are utilized as thickening, gel-forming, stabilizing, and active ingredient carriers [21]. Spirulina protein may be encapsulated by alginate, which makes it easier to create food with different protein sources [22]. Alginate is used in this ionic gelation encapsulation technology to trap many probiotics, improve their viability, and raise the likelihood that these microbes would survive adverse digestive tract circumstances: L. acidophilus [23,24], L. reuteri [25], L. casei [23,26,27], and Lactococcus lactis [28]. Alginate encapsulates containing linseed oil extract [29] and Satureja hortensis essential oil [30] were created using ionic gelation and have strong antioxidant qualities, making them suitable for use in pharmaceutical products. In another study [31], the curcumin encapsulation system was considered to have a high potential for the transport and controlled release of this bioactive”.

  1. The release of the essential oils from the capsules should be dependent on pH.  I recommend that the authors investigate the pH-dependent release.

Drug release from alginate encapsulates is dependent upon the penetrating of the dissolution medium into the spheres, the alginate matrix expanding and dissolving, and the active components dissolving via the enlarged matrix. The hydration and swelling that occur upon contact with the aqueous dissolution medium lead to the formation of a hydrogel surface layer that controls the influx of the aqueous media and the dissolution of the active components [88]. Osmotic or mechanical pressures, enzymes, and/or pH variations are some of the stimuli that cause changes in the gel phase, responsible for the release of the active components that are contained in the alginate matrix [41]. The in vitro dissolution characteristics of apigenin-loaded beads based on calcium alginate were examined in a recent work [88]. The release of the drug was delayed with an increase in sodium alginate content. The apparent cross-linking points inside the granules increased when the alginate content was raised, resulting in a stiff matrix and postponing the release of the active components. Within a day, the spheres released the assets.

In fact, the charge of the particles encasing essential oils varies with pH levels [89]. The degree of ionization of the various functional groups of the carriers [90] changes as a result. For instance, the release of bioactive chemicals from chitosan nanocapsules loaded with peppermint, green tea [91], Carum copticum [89], and carvacrol [92] was substantially higher at pH 3.0 than it was at higher pH values (pH between 7 and 11) [90].

In the present work, compared to the PBS medium (pH 7.4), the initial release reported in the study happens somewhat later in an acidic medium (pH 1.2). The low stability of these capsules in PBS medium serves as justification for this fact. About 58% of the eugenol in a PBS medium was released in 25 minutes. In contrast, this release was just 17% in an acidic medium. The release happened in a more regulated way and at comparable amounts in both media between 30 minutes and 2 hours. In an acidic media, almost 83% had been released after 3 hours, compared to roughly 63% in a PBS medium.

The microcapsules that were submerged in a pH 4.8 medium showed the highest rates of oregano essential oil release. Under these circumstances, high release levels at predefined times (78% in 7 hours) were noted. Lower release rates were observed in media with pH 4.2 (59% in 7 hours). Intermediate rates were observed in media with pH 3.8 [93]. According to Muneratto, Gallo, and Nicoletti, the complex weakened and release values increased at pH 4.8, where the zeta potential showed the smallest charge difference. The microcapsules were not destroyed, though. Given this, it can be suggested that the low initial release of eugenol in an acidic environment is related to the smaller difference in charge formed, thus being more stable.

We hope to have answered adequately your relevant comments.

Reviewer 3 Report

Comments and Suggestions for Authors

The paper “New insights from Syzygium aromaticum essential oil: encapsulation, characterization, and antioxidant activity” describes the ionic gelation with subsequent freeze-drying of clove essential oil. The paper seems to be interest because has a practical interest for the pharmaceutical sector, at the same time some questions  should be answered. 1) the authors should explain the phrase “New insights…’ from the title, because there is no explanation of this item through the text, or correct the title. 2) Is the composition of the clove oil that was used by the authors differ than that in the literature? Are these components are stable (Table 1)? 3) The authors should indicate if there any other approaches for encapsulation of clove essential oil in the literature. 4) the authors have explored the encapsulated CEO by different methods but no explanation of the choose of these methods is given 5) in Table 5 no reference is given 6) in discussion the text between lines 488-515 presents literature data about the activity of the main components of CEO but no correlation with the obtained data concerning the main point of Ms - encapsulation is suggested.

Author Response

  • Author's Reply to the Review Report (Reviewer 3)
  • Thanks for the careful analysis. We appreciated the comments.

The paper “New insights from Syzygium aromaticum essential oil: encapsulation, characterization, and antioxidant activity” describes the ionic gelation with subsequent freeze-drying of clove essential oil. The paper seems to be interest because has a practical interest for the pharmaceutical sector, at the same time some questions should be answered.

  1. The authors should explain the phrase “New insights…’ from the title, because there is no explanation of this item through the text, or correct the title.

The term “New insights” was used in the sense of “perspectives”.

Thanks, we removed the adjective New.

The title is now:

Insights from Syzygium aromaticum essential oil: encapsulation, characterization, and antioxidant activity

In the present work, we present promising data for the use of clove oil encapsulates by ionic gelation. Encapsulated clove oil is an alternative to its protection against factors such as high temperatures, low pressures, exposure to air and light, among others, which could contribute to the decomposition or evaporation of its active ingredients. Furthermore, this system can be used as a new asset delivery system. In the last paragraph of the introduction, we added this explanation of the title. We agreed with the suggestion and removed the “New”.

  1. Is the composition of the clove oil that was used by the authors differ than that in the literature? Are these components are stable (Table 1)?

No, the content of clove oil that is now in use is comparable to what has been documented in the literature. Nonetheless, there are differences in the amounts of each component in this oil. This deviation has been documented in the literature and is acceptable. Climate, soil, the way oil is extracted, and the portion of the plant that is utilized are some of the factors that cause this [64].

According to Kiki et al. (2023) [105], they are unstable chemicals that readily suffer oxidation and/or volatilization. Thus, it is recognized how crucial it is to safeguard these assets during the encapsulation process by employing a gentle procedure like ionic gelation.

 3) The authors should indicate if there any other approaches for encapsulation of clove essential oil in the literature.

Other methods of encapsulating clove oil include liposome systems, complex coacervation, and spray drying. The introduction's seventh paragraph now includes them.

The literature lists spray drying as one of the encapsulation techniques for clove oil [32-34]. Nevertheless, the last approach presents difficulties in encapsulating essential oils due to their volatility and sensitivity to high temperatures, potentially leading to the loss of significant components [35]. Specific equipment is needed, which raises expenses [36]. The encapsulated emulsion's deposition on the drying equipment's wall is another limiting issue that raises costs, lowers yield, and product quality [37]. Another popular technique is complex coacervation [38,39], however, it can be difficult to use on a large scale and lead to variances in the final product because it is easily agglomerated and offers little control over the size of the particles generated. In turn, encapsulation in liposomes necessitates the use of heating, which is problematic for the essential oil's stability, in addition to tools and solvents that, depending on the concentration employed, may be hazardous [40].

  1. The authors have explored the encapsulated CEO by different methods but no explanation of the choice of these methods is given.

Thanks for the excellent remarks.

In the penultimate paragraph of the introduction, the following text was inserted in regard to the various encapsulation techniques used in the current study:

"High temperatures or strong organic solvents are not necessary with ionic gelation, a mild encapsulation technique that helps maintain the integrity and pharmacological activity of the encapsulated drugs. It doesn't require any sophisticated and/or expensive equipment. This method, which makes use of biodegradable and biocompatible polymers, is fast and simple to repeat. High encapsulation efficiency enhanced chemical stability, and regulated release of active substances are all possible with ionic gelation [3,4,41,42]. However, as disadvantages, the generated encapsulated materials include a high proportion of water, which makes it easier for pathogenic microbes to contaminate them and lowers their thermal stability—two important aspects that lead to the product's degeneration. The goal of using freeze drying following ionic gelation is to address these problems [43]”.

  1. In Table 5 no reference is given.

Table 5 corresponds to data obtained by the authors through the DPPH (antioxidant) assay. However, paragraph 18 of the discussion cites the results of other authors, comparing them with those presented in this study.

 6. In discussion the text between lines 488-515 presents literature data about the activity of the main components of CEO but no correlation with the obtained data concerning the main point of Ms - encapsulation is suggested.

We have included the following paragraph to help you better comprehend the study: The EE% for LF1, LF2, and LF3 in the current work was 39.3% + 0.8, 50.4% + 0.6, and 76.9% + 0.5. Enhancing bioavailability and lowering dosages when used therapeutically are two benefits of high encapsulation rates, along with waste reduction and corresponding cost savings.

Thanks again.

Round 2

Reviewer 2 Report

Comments and Suggestions for Authors

The revised manuscript is acceptable.